# The H2S–Nrf2–Antioxidant Proteins Axis Protects Renal Tubular Epithelial Cells of the Native Hibernator Syrian Hamster from Reoxygenation-Induced Cell Death

**DOI:** 10.3390/biology8040074

**Published:** 2019-09-30

**Authors:** Theodoros Eleftheriadis, Georgios Pissas, Evdokia Nikolaou, Vassilios Liakopoulos, Ioannis Stefanidis

**Affiliations:** Department of Nephrology, Faculty of Medicine, University of Thessaly, Biopolis, Mezourlo Hill, 41110 Larissa, Greece; gpissas@msn.com (G.P.); nikolaoueyh@gmail.com (E.N.); liakopul@otenet.gr (V.L.); stefanid@med.uth.gr (I.S.)

**Keywords:** hibernation, H_2_S, Nrf2, oxidative stress, ischemia-reperfusion, cell death

## Abstract

During hibernation, repeated cycles of ischemia-reperfusion (I-R) leave vital organs without injury. Studying this phenomenon may reveal pathways applicable to improving outcomes in I-R injury-induced human diseases. We evaluated whether the H_2_S–nuclear factor erythroid 2-like 2 (Nrf2)–antioxidant proteins axis protects renal proximal tubular epithelial cells (RPTECs) of the native hibernator, the Syrian hamster, from reperfusion-induced cell death. To imitate I-R, the hamsters’, and control mice’s RPTECs were subjected to warm anoxia, washed, and then subjected to reoxygenation in fresh culture medium. Whenever required, the H_2_S-producing enzymes inhibitor aminooxyacetate or the lipid peroxidation inhibitor α-tocopherol were used. A handmade H_2_S detection methylene blue assay, a reactive oxygen species (ROS) detection kit, a LDH release cytotoxicity assay kit, and western blotting were used. Reoxygenation upregulated the H_2_S-producing enzymes cystathionine beta-synthase, cystathionine γ-lyase, and 3-mercaptopyruvate sulfurtransferase in the hamster, but not in mouse RPTECs. As a result, H_2_S production increased only in the hamster RPTECs under reoxygenation conditions. Nrf2 expression followed the alterations of H_2_S production leading to an enhanced level of the antioxidant enzymes superoxide dismutase 3 and glutathione reductase, and anti-ferroptotic proteins ferritin H and cystine-glutamate antiporter. The upregulated antioxidant enzymes and anti-ferroptotic proteins controlled ROS production and rescued hamster RPTECs from reoxygenation-induced, lipid peroxidation-mediated cell death. In conclusion, in RPTECs of the native hibernator Syrian hamster, reoxygenation activates the H2S–Nrf2–antioxidant proteins axis, which rescues cells from reoxygenation-induced cell death. Further studies may reveal that the therapeutic activation of this axis in non-hibernating species, including humans, may be beneficial in I-R injury-induced diseases.

## 1. Introduction

Ischemia-reperfusion (I-R) injury is implicated in many human diseases, from myocardial infarction and stroke to multiorgan failure [1,2,3]. Ischemia due to an occluded artery or decreased effective blood volume causes cell energy collapse, but even if blood supply is restored, reperfusion also induces cell injury through a burst of reactive oxygen species (ROS) production [1,2,3].

Hibernation is an adaptive mechanism encountered in many mammalian species in order to cope with the harshness of their environment. During hibernation periods of deep torpor, characterized by a robust decline in blood pressure, heart and breathing rates, and body temperature, are exchanged with short interbout arousals characterized by the restoration of the above parameters. Intriguingly, during hibernation, organs resist injury from the above cycles of I-R [4,5]. The discovery that in certain species, hibernation takes place with the preservation of high body temperature is significant, since, in human I-R injury, body temperature remains stable [6]. Finally, the identification of certain primate hibernators makes the study of this phenomenon extremely interesting, as these species are evolutionarily close to humans [7,8]. Clarifying how hibernators survive repeated cycles of I-R, may reveal new therapeutic strategies against I-R-induced human diseases.

The kidney is exceptionally vulnerable to I-R injury, since the metabolic demands of renal tubular cells are high, and not surprisingly, I-R injury is the most frequent cause of acute kidney injury (AKI) [9]. However, the kidneys of hibernators, such as the Syrian hamster (*Mesocricetus auratus*) and the dormouse (*Muscardinus avellanarius*), remain unaffected from the repeated cycles of cold I-R that characterize hibernation [10,11]. Interestingly, circulatory collapse due to bleeding or cardiac arrest followed by resuscitation, conditions that provoke warm I-R, induce severe renal impairment and pathological injury in the non-hibernating rat (*Rattus norvegicus domesticus)*, whereas the kidneys of the hibernator arctic ground squirrel (*Urocitellus parryii*) are saved [12]. Syrian hamster renal proximal tubular epithelial cells (RPTECs) resist warm anoxia-reoxygenation injury as well. On the contrary, mouse (*Mus musculus*) RPTECs die from apoptosis during warm anoxia and ferroptosis during reoxygenation [13]. Yet, the pathways involved in the protection of hamster RPTECs from the oxidative stress and the subsequent ferroptosis during reoxygenation are not well-understood. 

In this study, we evaluated whether the H_2_S-nuclear factor erythroid 2-like 2 (Nrf2)-antioxidant proteins axis protects the RPTECs of the native hibernator Syrian hamster from cell death due to the reoxygenation that follows warm anoxia. For this purpose, we assessed the levels of the H_2_S producing enzymes cystathionine beta-synthase (CBS), cystathionine γ-lyase (CSE), and 3-mercaptopyruvate sulfurtransferase (3-MST) in reoxygenated hamster RPTECs, and the production of H_2_S [14]. Then, we examined whether the accumulated H_2_S increases the Nrf2 level, since this gas alters the conformation of the Kelch-like ECH-associated protein 1 (Keap1), releasing the Nrf2 bound to it and sparing the latter from proteasomal degradation [15,16]. Nrf2 translocates into the nucleus, binds to promoters containing the antioxidant response element (ARE), and transcribes many antioxidant genes [17,18]. Thus, we check whether the upregulated Nrf2 level enhances the expression of representative antioxidant proteins, which are known to be under Nrf2’s transcriptional control. 

To verify the significance of the H2S–Nrf2–antioxidant proteins axis on antioxidant enzymes and cell survival during reoxygenation, we also assessed the impact of the non-specific inhibitor of H_2_S-producing enzymes aminooxyacetate (AOAA) on H_2_S and ROS production, as well as on cell survival. AOAA inhibits CBS and CSE [19]. Additionally, by inhibiting transamination, AOAA is expected to inhibit 3-MST as well, since 3-MST catalyzes the conversion of cysteine to pyruvate with the assistance of cysteine aminotransferase in a two-step reaction [20]. The effect of α-tocopherol, a known inhibitor of lipid peroxidation and ferroptosis [21], alone or in combination with AOAA, on cell survival, was also evaluated. For comparison, the same parameters were assessed in RPTECs of the non-hibernator mouse. 

## 2. Materials and Methods

### 2.1. Cells and Culture Conditions

Primary Syrian hamster (catalogue number HM-6015, Cell Biologics, Chicago, IL, USA) and C57BL/6 mouse RPTECs (catalogue number C57-6015, Cell Biologics) were cultured in a Complete Epithelial Cell Medium kit, and supplemented with epithelial cell growth supplement, fetal bovine serum, and antibiotics (catalogue number M6621, Cell Biologics). According to the manufacturer’s instructions, the contents of the Epithelial Cell Medium supplement kit containing 0.5 mL insulin-transferrin-selenium, 0.5 mL epithelial growth factor, 5 mL L-glutamine, 5 mL antibiotic-antimycotic solution, and 10 mL fetal bovine serum, were added to 500mL of basal medium before use. Cells were expanded in 75 cm^2^ flasks and passage two cells were used for the experiments.

Cells were cultured in 96-well plates (10^4^ cells/well) or 6-well plates (3 × 10^5^ cells/well) for 12 h before the onset of anoxia at 37 °C. Cell confluency detected by inverted microscopy did not differ at the start of each experiment. The GasPak^TM^ EZ Anaerobe Container System with Indicator (catalogue number 26001, BD Biosciences, S. Plainfield, NJ, USA) was used to decrease oxygen concentration to less than 1%. These anoxic conditions simulated warm ischemia and lasted for 24 h.

Next, cells were washed with Dulbecco's phosphate buffer saline (PBS) (Sigma-Aldrich; Merck Millipore, Darmstadt, Germany), the culture medium was replenished, and the cells were cultured in a humidified atmosphere containing 5% CO_2_ at 37 °C. These reoxygenation conditions imitated warm reperfusion and lasted for 2 h.

The periods of anoxia and reoxygenation were selected in accordance with a previous study of our laboratory, upon which it was shown that primary Syrian hamster RPTECs are extremely resistant to both anoxia and reoxygenation, whereas primary mouse RPTECs decline considerably after 48 h of anoxia or 4 h of reoxygenation [13]. All the experiments were repeated nine times.

Whenever used, the H_2_S-producing enzymes inhibitor AOAA (Selleck Chemicals, Munich, Germany) at a concentration of 2 mM, was added. This concentration was selected after assessing the cytotoxicity of AOAA in RPTECs (see below). Additionally, when required, the inhibitor of lipid peroxidation and ferroptosis α-tocopherol (Sigma-Aldrich; Merck Millipore) was used at a concentration of 100 μM.

### 2.2. Assessment of AOAA Toxicity in RPTECs

Since the concentrations of AOAA vary in different studies, we did preliminary experiments to determine a non-toxic AOAA concentration for RPTECs. For this purpose, hamster and mouse RPTECs were cultured for 26 hours in 96-well plates in a humidified atmosphere containing 5% CO_2_ in the presence or not of escalated concentrations of AOAA (0.5, 1, and 2 mM). Cytotoxicity was assessed with LDH release assay using the Cytotox Non-Radioactive Cytotoxic Assay kit (Promega Corporation, Madison, WI, USA). Cell death was calculated by the equation Cell death (%) = (LDH in the supernatant: Total LDH) × 100. These experiments were performed in triplicates and repeated nine times. As depicted in Figure 1, AOAA was not cytotoxic at concentrations of 0.5, 1, or 2 mM. The latter concentration of 2 mM was selected for all subsequent experiment.

### 2.3. Assessment of Proteins of Interest

Hamster and mouse RPTECs were cultured in 6-well plates as previously described. Once the reoxygenation period was over, cells were lysed with the T-PER tissue protein extraction reagent (Thermo Fisher Scientific Inc., Waltham, MA, USA) supplemented with protease and phosphatase inhibitors (Sigma-Aldrich; Merck Millipore and Roche Diagnostics, Indianapolis, IN, USA, respectively). Protein was quantified via Bradford assay (Sigma-Aldrich; Merck Millipore) and 10 μg from each sample were used for western blotting.

For western blotting, 4–12% bis-tris acrylamide gels (NuPAGE 4–12% Bis-Tris Gel 1.0 mm × 15 well, Invitrogen; Thermo Fisher Scientific, Inc.) were used, and polyvinylidene difluo­ride (PVDF) membrane blots were incubated with the primary antibody for 16 h at 4 °C, and then with the secondary antibody for 30 min at room temperature. Whenever reprobing of the PVDF blots was required, the Restore Western Blot Stripping Buffer (Thermo Fisher Scientific Inc.) was used.

It should be noted that mouse and hamster proteins were electrophoresed in different gels. This was done on purpose, since even slight differences in the structure of a protein derived from different species can affect the affinity of the antibody used for the western blotting significantly. Thus, even if both mouse and hamster proteins were electrophoresed in the same gel, the direct comparison between them would be inaccurate. 

Primary antibodies were rabbit polyclonal antibody against CBS (dilution: 1:1000, catalogue number TA338394, OriGene Technologies Inc., Rockville, MD, USA), mouse monoclonal against CSE (dilution: 1:100, catalogue number sc-374249, Santa Cruz Biotechnology, Dallas, TX, USA), mouse monoclonal antibody against 3-MST (dilution: 1:100, catalogue number sc-376168, Santa Cruz Biotechnology), rabbit polyclonal antibody against Nrf2 (dilution: 1:1000, catalogue number TA343586, OriGene Technologies), mouse monoclonal antibody against superoxide dismutase 3 (SOD3) (dilution: 1:100, catalogue number sc-271170, Santa Cruz Biotechnology), mouse monoclonal antibody against glutathione reductase (GR) (dilution: 1:100, catalogue number sc-133245, Santa Cruz Biotechnology), mouse monoclonal antibody against the ferritin heavy (H) chain (dilution: 1:100, catalogue number sc-376594, Santa Cruz Biotechnology), rabbit polyclonal antibody against cystine-glutamate antiporter (xCT) (dilution: 1:1000, catalogue number ANT-111, Alomone Labs, Jerusalem, Israel), and rabbit polyclonal antibody against β-actin (dilution: 1:2500, catalogue number 4967, Cell Signaling Technology, Cell Signaling Technology, Danvers, MA, USA). Anti-mouse IgG, HRP-linked antibody (dilution: 1:1000, catalogue number 7076, Cell Signaling Technology) or anti-rabbit IgG, HRP-linked antibody (dilution: 1:1000, catalogue number 7074, Cell Signaling Technology) were used as secondary antibodies.

The LumiSensor Plus Chemiluminescent HRP Substrate kit (GenScript Corporation, Piscataway, NJ, USA) was used for enhanced chemiluminescent detection of the western blot bands, and the Image J software (National Institute of Health, Bethesda, MD, USA) for their densitometric analysis. These experiments were repeated nine times. (Whole western blots are available in Materials: File Appendix A).

### 2.4. Measurement of H_2_S Production

At the end of the 2-h reoxygenation period, H_2_S production was assessed by measuring its concentration in the supernatants of RPTECs cultured in 6-well plates under the previously noted conditions. For this purpose, an already described methylene blue assay was performed with some modifications [22,23]. To trap the produced H_2_S, zinc acetate (1% w/v) (Sigma-Aldrich; Merck Millipore) was added immediately to 1 mL of each supernatant. In order to prepare the required diamine-ferric solution, 400 mg N,N-dimethyl-p-phenylenediamine dihydrochloride (Sigma-Aldrich; Merck Millipore) were dissolved in 10 mL of 6 M HCl, 600 mg ferric chloride (Sigma-Aldrich; Merck Millipore) in 10 mL 6 M HCl, and then, 1 mL from each of the two solutions were mixed. Next, 50 μL of the diamine-ferric solution was added to each supernatant for a 30 min incubation at 37 °C. Once the incubation period was over, 200 μL from each reaction was seeded in a 96-well plate, and the amount of methylene blue formed in each supernatant was measured at a wavelength of 670 nm on an EnSpire® Multimode Plate Reader (PerkinElmer, Waltham, MA, USA). Values derived from measurements in wells without cells (blank) were subtracted. Different concentrations of methylene blue (Merck, Burlington, MA, USA) were also prepared to create the necessary standard curve for the measurement of methylene blue produced by each experimental reaction. These experiments were performed in triplicates and repeated nine times.

### 2.5. Assessment of ROS Production

ROS production was assessed in RPTECs cultured in 96-well plates under the previously described conditions. Once reoxygenation period was over, cells were stained with 5 μM of the fluorogenic probe CellROX® Deep Red Reagent (Invitrogen, Life Technologies, Carlsbad, CA, USA) by adding the probe to the culture medium and incubating the cells at 37 °C for 30 min. Then, RPTECs were washed with PBS, and fluorescence signal intensity was measured on an EnSpire® Multimode Plate Reader. These experiments were performed in triplicates and repeated nine times.

### 2.6. Evaluation of Cell Death

RPTECs were cultured in 96-well plates under the conditions described above. At the end of the reoxygenation period cell, death was assessed with LDH release assay using the Cytotox Non-Radioactive Cytotoxic Assay kit. Cell death was calculated by the equation cell death (%) = (LDH in the supernatant: Total LDH) × 100. These experiments were performed in triplicates and repeated nine times.

### 2.7. Statistical Analysis

Data were analyzed with the IBM SPSS Statistics for Windows, version 20 (IBM Corp., Armonk, NY, USA). One-sample Kolmogorov–Smirnov test was used for testing and confirming that the evaluated variables were normally distributed. For comparisons of means, an unpaired *t*-test or one-way analysis of variance followed by Bonferroni’s correction test were used. Results were expressed and depicted as means ± standard errors of means (SEMs) and a *p* < 0.05 was considered statistically significant. For analysis of the western blotting results, the optical densities of the bands were used. However, for the reader’s convenience, they are presented after normalization for the control group of untreated cells.

## 3. Results

### 3.1. Reoxygenation Increases the Level of H_2_S-Producing Enzymes in the Hamster, but Not in Mouse RPRECs

Hamster and mouse RPTECs were subjected to 24 hours of warm anoxia, washed with PBS, and then subjected to 2 hours’ reoxygenation in fresh culture medium. The levels of H_2_S-producing enzymes CBS, CSE, and 3-MST after the 2 h of reoxygenation, compared to hamster RPTECs not subjected to anoxia and reoxygenation, increased by factors of 2.78 ± 0.26, 2.10 ± 0.22, and 1.64 ± 0.12, respectively (*p* < 0.001 in all cases). On the contrary, the levels of CBS, CSE, and 3-MST did not change significantly in mouse RPTECs (Figure 2). 

### 3.2. Reoxygenation Increases H_2_S Production in a H_2_S-Producing Enzymes-Dependent Way in the Hamster, but Not in Mouse RPTECs

Hamster and mouse RPTECs were cultured in the presence or not of 2 mM AOAA. Cells were subjected to 24 h warm anoxia, washed with PBS, and then subjected to 2 h reoxygenation in fresh culture medium. Production of H_2_S was assessed by its concentration in the supernatants. Reoxygenation increased H_2_S production only in hamster RPTECs (34.78 ± 0.86 μM versus 12.44 ± 0.38 μM, *p* < 0.001), while it left H_2_S production unaffected in mouse RPTECs (10.44 ± 0.41 μM versus 9.44 ± 0.29 μM, *p* n.s.). Whenever applied, the inhibitor of the H_2_S-producing enzymes AOAA decreased H_2_S production (Figure 3A).

### 3.3. Reoxygenation Increases the Nrf2 Level in a H_2_S-Producing Enzymes-Dependent Way in the Hamster, but Not in Mouse RPTECs

Hamster and mouse RPTECs were cultured with or without the H_2_S-producing enzymes inhibitor AOAA (2 mM). Cells were subjected to 24 hours’ warm anoxia, washed with PBS, and then subjected to 2 h reoxygenation in fresh culture medium. Cell protein was extracted, and the level of Nrf2 was assessed by western blotting. The changes in Nrf2 level followed the changes in H_2_S concentration. Levels of Nrf2 after the 2 h of reoxygenation increased by a factor of 1.90 ± 0.05 (*p* < 0.001), compared to hamster RPTECs not subjected to anoxia and reoxygenation. On the contrary, the level of Nrf2 did not alter significantly in mouse RPTECs. Whenever applied, AOAA decreased Nrf2 level (Figure 3B,C).

### 3.4. Reoxygenation Increases the Expression of SOD3, GR, Ferritin H, and xCT in the Hamster, but Not in Mouse RPTECs

Hamster and mouse RPTECs were subjected to 24 h warm anoxia, washed with PBS, and then subjected to 2 h reoxygenation in fresh culture medium. Cell protein was extracted, and the expression of SOD3, GR, ferritin H, and xCT were assessed by western blotting. The changes in expression of the evaluated proteins followed same trend with the changes observed with Nrf2. Expression of SOD3, GR, ferritin H, and xCT after 2 h of reoxygenation, compared to hamster RPTECs not subjected to anoxia and reoxygenation, increased by a factor of 2.10 ± 0.22, 3.70 ± 0.52, 2.84 ± 0.22, and 1.74 ± 0.13, respectively (*p* < 0.001 in all cases). Again, the expression of SOD3, GR, ferritin H, and xCT did not alter significantly in mouse RPTECs (Figure 4). 

### 3.5. Under Reoxygenation, the H_2_S-Producing Enzymes Play a Significant Role in Controlling ROS Production in Both Hamster and Mouse Cells

Hamster and mouse RPTECs were cultured with or without 2 mM AOAA. ROS production was assessed in cells subjected to 24 h warm anoxia, washed with PBS and then subjected to 2 h reoxygenation in fresh culture medium. Compared to hamster RPTECs not subjected to anoxia and reoxygenation, reoxygenation induced ROS production in both hamster RPTECs (signal intensity 61.56 ± 1.74 versus 42.00 ± 0.50, *p* < 0.001) and mouse RPTECs (signal intensity 100.11 ± 0.89 versus 43.00 ± 0.76, *p* < 0.001). Inhibition of H_2_S-producing enzymes with AOAA increased ROS production even further in both hamster RPTECs (signal intensity 97.89 ± 1.44 versus 61.56 ± 1.74, *p* < 0.001) and mouse RPTECs (signal intensity 120.78 ± 2.09 versus 97.89 ± 1.44, *p* < 0.001) (Figure 5A).

### 3.6. H_2_S-Producing Enzymes Rescue Hamster RPTECs from Reoxygenation-Induced, Lipid Peroxidation-Mediated Cell Death and Also Decrease Reoxygenation Cytotoxicity in Mouse RPTECs

Hamster and mouse RPTECs were cultured with or without the H_2_S-producing enzymes inhibitor AOAA (2 mM), and the lipid peroxidation inhibitor α-tocopherol (100 μM). Cell death was assessed in cells subjected to 24 h warm anoxia, washed with PBS, and then subjected to 2 hours’ reoxygenation in fresh culture medium. In hamster RPTECs, reoxygenation did not induce cell death (cytotoxicity 9.33% ± 0.30% versus 9.39% ± 0.21%, p n.s.). However, in the presence of AOAA, reoxygenation caused cell death (cytotoxicity 24.00% ± 0.37% versus 9.39% ± 0.21%, *p* < 0.001), which was ameliorated when α-tocopherol was also present (cytotoxicity 14.00% ± 0.33% versus 24.00% ± 0.37%, *p* < 0.001). In mouse RPTECs, reoxygenation caused cell death (cytotoxicity 31.56% ± 0.47% versus 9.89% ± 0.20%, *p* < 0.001). The presence of AOAA aggravated reoxygenation-induced cell death (cytotoxicity 38.11% ± 0.59% versus 31.56% ± 0.47%, *p* < 0.001). The lipid peroxidation inhibitor α-tocopherol ameliorated reoxygenation-induced cell death both in the absence (cytotoxicity 17.11 ± 0.35%, *p* < 0.001) and the presence of AOAA (cytotoxicity 22.00 ± 0.67% *p* < 0.001) (Figure 5B).

## 4. Discussion

During hibernation, repeated cycles of I-R leave behind vital organs without injury [4,5]. Studying this phenomenon may reveal pathways applicable to the prevention or amelioration of I-R injury-induced human diseases. The discovery of certain primate hibernators supports such an option since these species are evolutionary close to humans [7,8].

The increased ROS production during the reperfusion phase is one of the major causes of cells and organs injury [1,2,3]. In this study, we evaluated how hibernators escape from the reperfusion-induced cell damage. For this purpose, we cultured RPTECs derived from the native hibernator, the Syrian hamster, and the non-hibernator mouse, since, at least in non-hibernating species, RPTECs are incredibly vulnerable to I-R injury [9]. To simulate the I-R conditions, RPTECs were subjected to warm anoxia, washed with PBS, and then subjected to reoxygenation in fresh culture medium. Then we assessed the H2S–Nrf2–antioxidant proteins pathway. This pathway, albeit not well-studied in hibernators, defends against ROS disruption of cell integrity [18]. We evaluated the H2S–Nrf2–antioxidant proteins axis only under reoxygenation conditions, since according to a previous study of our laboratory [13], both in mouse and hamster RPTECs, ROS production increases under reoxygenation, when oxygen is available, and not under anoxia. Thus, this axis could play a protective role only under reoxygenation. CBS, CSE, and 3-MST are the three enzymes that produce H_2_S [14]. Once H_2_S increases, it alters the conformation of Keap1, releasing the bound to it Nrf2 and sparing the latter from proteasomal degradation [15,16]. The accumulated Nrf2 translocates into the nucleus, get phosphorylated, binds to ARE containing promoters, and transcribes many antioxidant genes [17,18]. We evaluated the total Nrf2 saved from proteasomal degradation. Instead of assessing nuclear or phosphorylated Nrf2, we evaluated the level of the proteins SOD3, GR, ferritin H, and xCT, whose genes are known to be under the transcriptional control of Nrf2 [17]. 

We found that in the hamster but not mouse RPTECs, reoxygenation induces the expression of all H_2_S-producing enzymes. In hamster RPTECs, this increase in the level of H_2_S-producing enzymes is accompanied by enhanced H_2_S production and Nrf2 expression. The administration of AOAA, an H_2_S-producing enzymes inhibitor [19,20], inhibited both H_2_S production and Nrf2 upregulation, signifying the role of these enzymes in the latter outcomes. Interestingly, AOAA reduced H_2_S production and Nrf2 expression in both hamster and mouse RPTECs, though the latter cells did not exhibit any enhanced H_2_S production and Nrf2 level under reoxygenation. 

Next, we evaluated whether increased Nrf2 boosts the expression of representative antioxidant proteins, known to have ARE within their gene promoter [17]. The antioxidant enzymes SOD3 and GR were upregulated significantly in hamster RPTECs subjected to reoxygenation, but not in mouse RPTECs. SOD catalyzes the dismutation of the superoxide radical into either molecular oxygen or hydrogen peroxide [24]. GR catalyzes the reduction of glutathione disulfide to the sulfhydryl form of glutathione, which is a critical molecule in resisting oxidative stress [24].

In addition to antioxidant enzymes, Nrf2 also upregulates proteins that protect cells from ROS-induced lipid-peroxidation-mediated ferroptosis [17]. Accordingly, we detected that in the hamster, but not in mouse RPTECs, reoxygenation upregulates ferritin and xCT. Ferritin, by sequestrating intracellular labile iron, prevents lipid peroxidation through the Fenton reaction or iron-containing lipoxygenases [21]. xCT, by facilitating entry of cystine into the cells, offers the necessary building blocks for glutathione production [21]. 

To verify the role of the H2S–Nrf2–antioxidant enzymes axis activated by reoxygenation, in controlling oxidative stress in RPTECs, we assessed the impact of the H_2_S-producing enzymes inhibitor AOAA on ROS production. Expectably, we found that AOAA increased ROS in hamster RPTECs subjected to reoxygenation. Interestingly, the same was observed in mouse RPTECs. The latter indicates that the evaluated axis plays a significant role in controlling oxidative stress in both species. However, the H2S–Nrf2–antioxidant enzymes axis is activated by reoxygenation only in hamster RPTECs.

Ferroptosis or membrane lipid peroxidation-induced cell death plays an essential role in I-R ROS-mediated cell death [21]. In non-hibernating species, ferroptosis has been identified as a type of renal tubular epithelial cell death due to I-R and the cause of subsequent AKI [25,26,27,28,29]. Various other types of cell death have also been incriminated for I-R-induced AKI [30]. In one in vivo study, necroptosis, a type of cell death that requires activation of the immune system [30], prevailed in the mouse kidneys subjected to I-R [31]. However, the same investigators confirmed later that in isolated mouse renal tubules subjected to anoxia-reoxygenation, reoxygenation-induced cell death occurred through ferroptosis [25]. Hence, it is likely that ferroptosis is the initial type of cell death due to reoxygenation. Next, danger associated molecular patterns are released by the necrotic cells and activate the immune system [32], resulting in a secondary wave of necroptotic cell death. Importantly, cultured Syrian hamster RPTECs resist warm anoxia-reoxygenation injury, whereas mouse RPTECs die from apoptosis during warm anoxia and ferroptosis during reoxygenation [13]. To evaluate the role of the H_2_S-Nrf2-anti-ferroptotic proteins axis, activated by reoxygenation, in hamster RPTECs, we assessed the impact of the H_2_S-producing enzymes inhibitor AOAA and the lipid-peroxidation inhibitor α-tocopherol on cell death. According to our results, activation of the former pathway rescues hamster RPTECs from reoxygenation-induced lipid peroxidation-mediated cell death. Although this axis was not activated by reoxygenation in mouse RPTECs, AOAA also increased lipid peroxidation-induced cell death in these cells. However, since the activity of the H2S–Nrf2–antioxidant proteins axis was not intensified by reoxygenation in mouse RPTECs, these cells were not rescued. 

Several elements of the H2S–Nrf2–antioxidant proteins axis have been assessed in hibernating species in previous studies [10,23,33,34,35,36,37]; however, to our knowledge, this is the first time that all the components of this axis were evaluated mechanistically. Additionally, and contrary to previous studies [10,23,33,34,35,36,37], we assessed the activity and the outcome of the H2S–Nrf2–antioxidant proteins axis in a native hibernator, not under cold anoxia-reoxygenation conditions, but under warm anoxia-reoxygenation conditions, such are the conditions encountered in clinical practice. However, the available data about the protective role of H_2_S in hamster renal cells subjected to cold anoxia-reoxygenation may be proven useful in the field of solid organ transplantation, and more precisely, in graft preservation [10,38].

Our results confirmed that, contrary to the RPTECs of the non-hibernator mouse, the H2S–Nrf2–antioxidant proteins axis is activated by reoxygenation in Syrian hamster RPTECs and protects them from cell death. The reoxygenation-induced upregulation of the H_2_S-producing enzymes launches this pathway. Clarifying the mechanisms involved in CBS, CSE, and 3-MST upregulation in hamster RPTECs may reveal new therapeutic strategies for improving the outcome of I-R injury in human diseases. Another option would be the administration of H_2_S donors. In non-hibernating species, this approach has been tested in experimental models of AKI [39,40,41], myocardial infarction [42], cerebral stroke [43], and multiorgan failure [44,45], with promising results. The development and trial of Nrf2 activators other than H_2_S donors may also be proven useful, since, in experimental studies, such substances improved the course of I-R-induced AKI in mice [46,47]. Our research supports the above strategies, since the H2S–Nrf2–antioxidant protein axis was detected to be upregulated in reoxygenated Syrian hamster RPTECs and rescued them from lipid peroxidation-mediated cell death. 

Nevertheless, one should keep in mind that in I-R injury, cell death occurs not only during the reoxygenation period but also during the preceding anoxia. For example, mouse RPTECs die through apoptosis during warm anoxia and ferroptosis during reoxygenation [13]. Thus, a combination of substances that inhibit different types of cell death, for instance, apoptosis and ferroptosis, may be required for an optimum effect.

A limitation of our study is its in vitro nature, since direct conclusions cannot always be extrapolated safely from the in vitro to the in vivo model. Nevertheless, this study could be considered as a starting point for further research aiming to define the molecular mechanisms involved in the resistance of hibernating species cells to warm I-R-injury. Moreover, fields under investigation in vitro studies, through their strictly controlled conditions, may help in the detection of the precise sequence of the events that occur in vivo. An excellent example is the already noted paradigm of the sequence of reoxygenation-induced ferroptosis, ferroptosis-induced activation of the immune system, and activated immune system-induced necroptosis [25,30]. 

## 5. Conclusions

In conclusion, in RPTECs of the native hibernator Syrian hamster, reoxygenation activates the H2S–Nrf2–antioxidant proteins axis, which rescues cells from reoxygenation-induced cell death. Further studies may reveal that the therapeutic activation of this axis in non-hibernating species, including humans, may be beneficial in I-R injury-induced diseases. 

## Figures and Tables

**Figure 1 biology-08-00074-f001:**
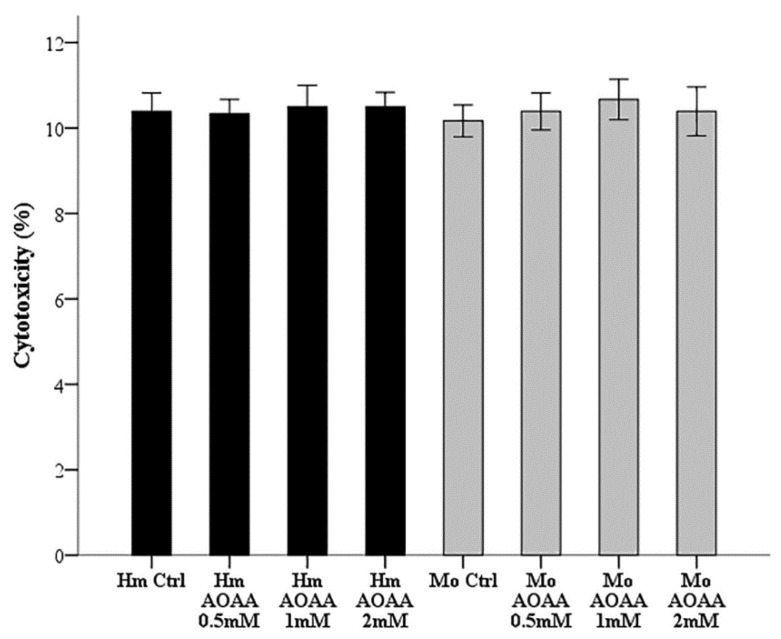
Assessment of AOAA toxicity in renal proximal tubular epithelial cells (RPTECs). Hamster and mouse RPTECs were cultured for 26 h in a humidified atmosphere containing 5% CO*_2_* without or with escalated concentrations of AOAA (0.5, 1, and 2 mM). In all the evaluated concentrations, AOAA was not toxic for either hamster or mouse RPTECs. Error bars correspond to standard error of mean (SEM).

**Figure 2 biology-08-00074-f002:**
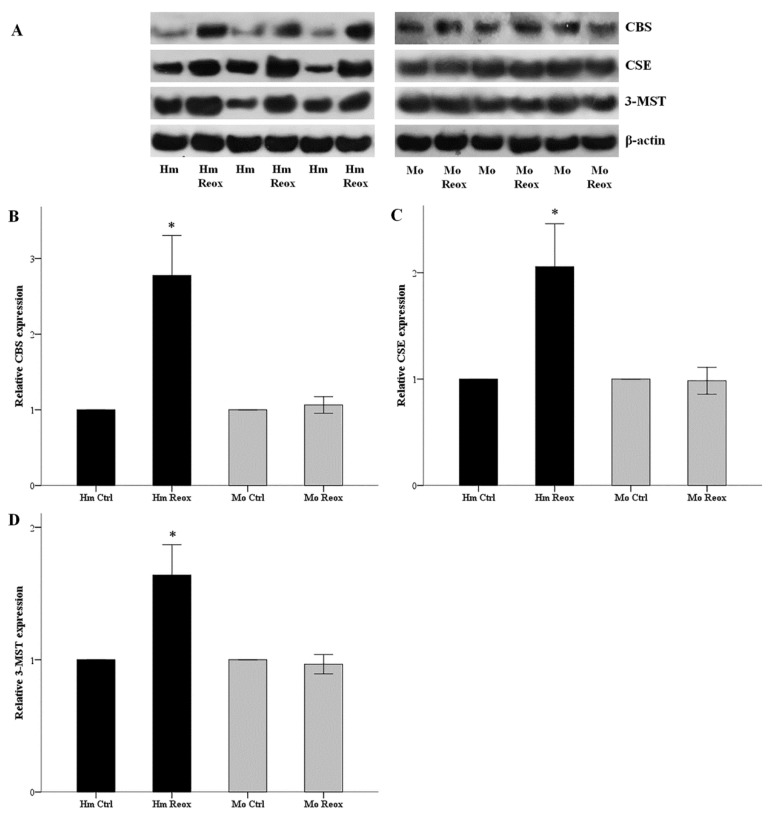
Reoxygenation increases the level of H_2_S-producing enzymes in the hamster, but not in mouse RPRECs. Hamster and mouse RPTECs were subjected to 24 h warm anoxia, washed with PBS, and then subjected to 2 h reoxygenation in fresh culture medium. Protein was extracted for western blotting. Panel (**A**) depicts three of the nine performed experiments. Compared to hamster RPTECs not subjected to anoxia and reoxygenation, the level of the H_2_S-producing enzymes CBS, CSE, and 3-MST after the 2 hours of reoxygenation increased significantly. On the contrary, the level of CBS, CSE, and 3-MST did not alter significantly in mouse RPTECs (**B,C,D**). An asterisk corresponds to a *p* < 0.001 compared to the control cells, and error bars correspond to SEM.

**Figure 3 biology-08-00074-f003:**
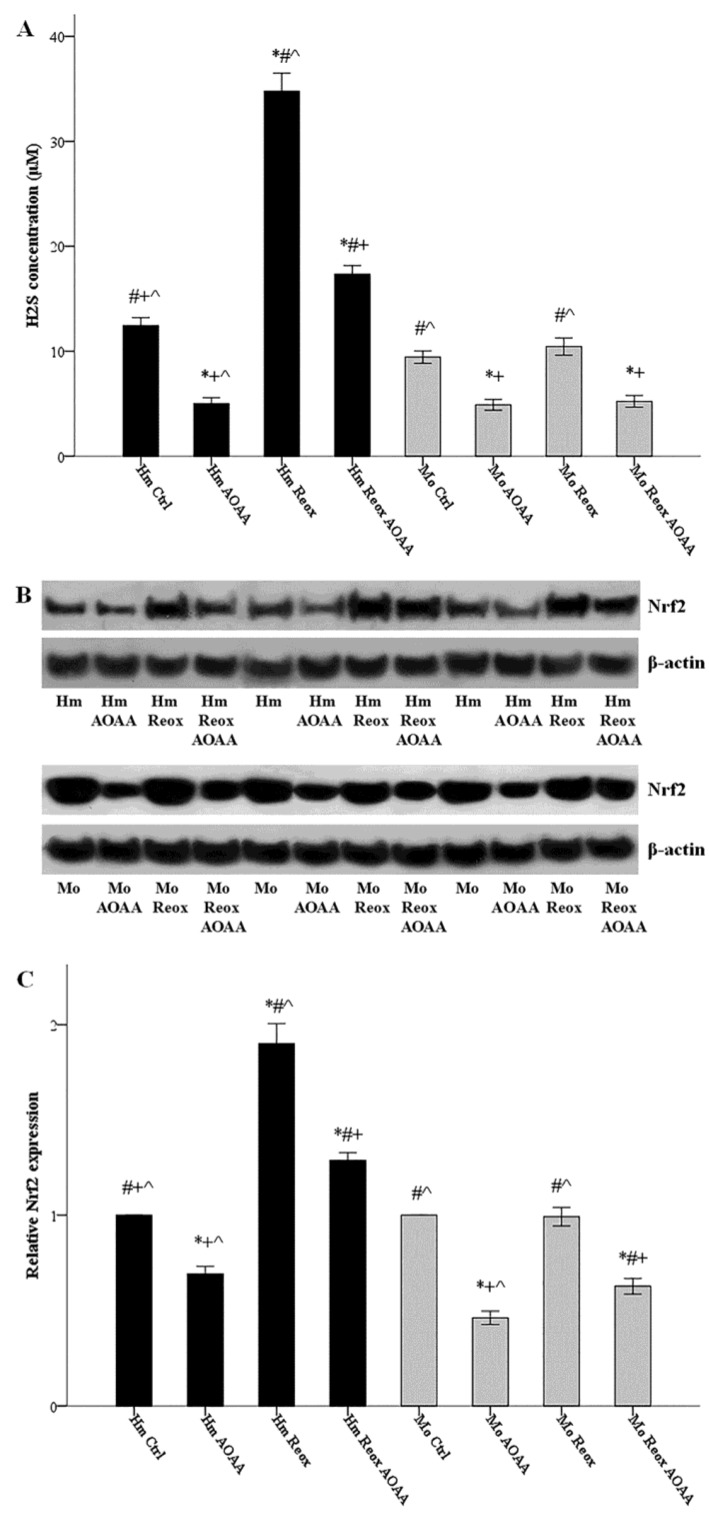
Reoxygenation increases H_2_S production and Nrf2 levels in a H_2_S-producing enzymes dependent way in the hamster, but not in mouse RPTECs. Hamster and mouse RPTECs were cultured in the presence or not of the H_2_S-producing enzymes inhibitor AOAA (2 mM). Cells were subjected to 24 h warm anoxia, washed with PBS, and then subjected to 2 h reoxygenation in fresh culture medium. Production of H_2_S was assessed by its concentration in the supernatants. Reoxygenation increased H_2_S production only in hamster RPTECs, while it left H_2_S production unaffected in the mouse. Whenever applied, AOAA reduced H_2_S production (**A**). Nrf2 was evaluated with western blotting, and three of the nine performed experiments are depicted in panel (**B**). The changes in Nrf2 level followed the changes in H_2_S concentration. Compared to hamster RPTECs not subjected to anoxia and reoxygenation, the level of Nrf2 after the 2 h of reoxygenation increased significantly. On the contrary, the level of Nrf2 did not change in mouse RPTECs. Whenever applied, AOAA decreased Nrf2 level (**C**). Symbols *, #, + and ^ correspond to a *p* < 0.001 compared to the first, second, third, and fourth depicted conditions, respectively. Error bars correspond to SEM.

**Figure 4 biology-08-00074-f004:**
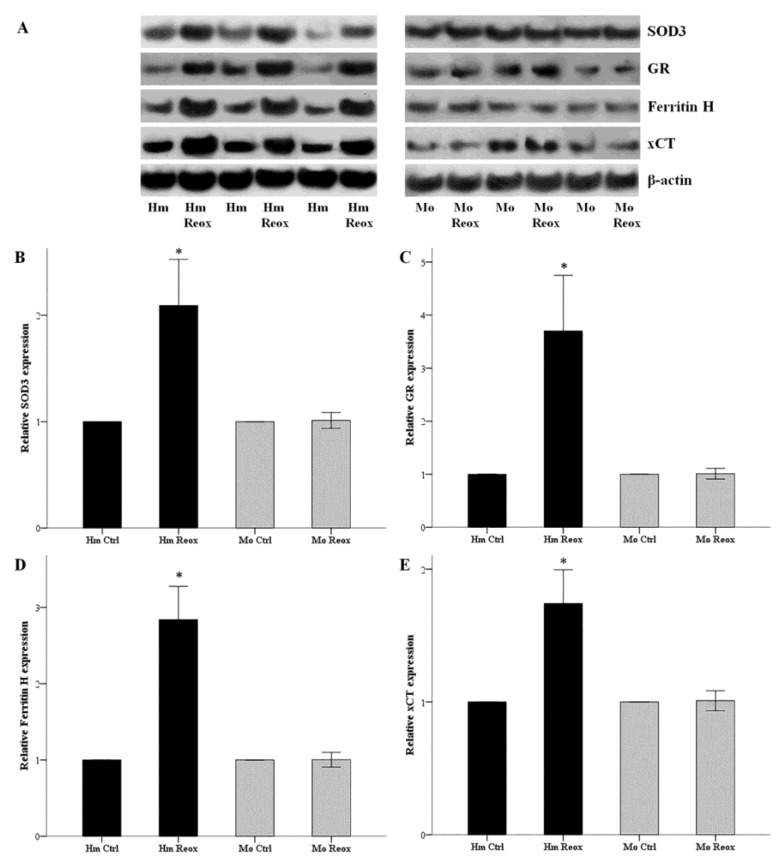
Reoxygenation increases the expression of SOD3, GR, ferritin H and xCT in the hamster, but not in mouse RPTECs. Hamster and mouse RPTECs were subjected to 24 h warm anoxia, washed with PBS, and then subjected to 2 h reoxygenation in fresh culture medium. Cell protein was extracted, and the expression of SOD3, GR, ferritin H, and xCT were assessed by western blotting. Panel A depicts three of the nine performed experiments. The changes in expression of the evaluated proteins followed the changes in Nrf2 levels. Compared to hamster RPTECs not subjected to anoxia and reoxygenation, the levels of SOD3, GR, ferritin H, and xCT after the 2 h of reoxygenation increased considerably. On the contrary, the expressions of SOD3, GR, ferritin H, and xCT did not alter significantly in mouse RPTECs (**B,C,D**,**E**). An asterisk corresponds to a *p* < 0.001 compared to the control cells, and error bars to SEM.

**Figure 5 biology-08-00074-f005:**
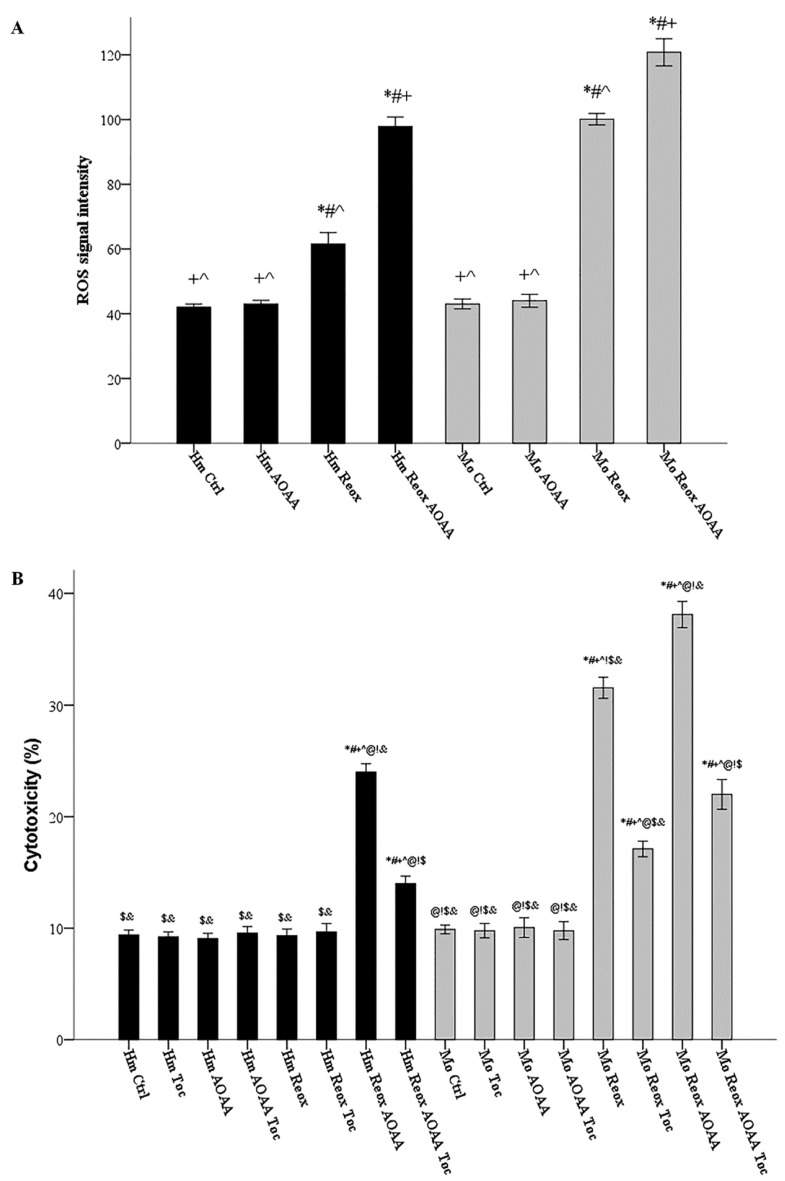
Under reoxygenation, the H_2_S-producing enzymes control ROS production in both hamster and mouse cells; they rescue hamster RPTECs from reoxygenation-induced lipid peroxidation-mediated cell death and also decrease reoxygenation cytotoxicity in mouse RPTECs. Hamster and mouse RPTECs were cultured with or without the H_2_S-producing enzymes inhibitor AOAA (2 mM), and the lipid peroxidation inhibitor α-tocopherol (100 μM). Cells were subjected to 24 hours’ warm anoxia, washed with PBS, and then subjected to 2 h reoxygenation in fresh culture medium. Compared to hamster RPTECs not subjected to anoxia and reoxygenation, reoxygenation induced ROS production in both hamster and mouse RPTECs. AOAA increased ROS production further in both hamster and mouse RPTECs (**A**). In hamster RPTECs, reoxygenation did not induce cell death. However, in the presence of AOAA, reoxygenation caused cell death, which was ameliorated when α-tocopherol was also present. In mouse RPTECs, reoxygenation caused cell death. The presence of AOAA aggravated reoxygenation-induced cell death, while α-tocopherol ameliorated reoxygenation-induced cell death both in the absence and presence of AOAA (**B**). Symbols *, #, +, ^, @, !, $, and & correspond to a *p* < 0.001 compared to the first, second, third, fourth, fifth, sixth, seventh, and eighth depicted conditions, respectively. Error bars correspond to SEM.

## Data Availability

The analyzed datasets generated during the study are available from the corresponding author on reasonable request.

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
