# Peer review of "The H2S–Nrf2–Antioxidant Proteins Axis Protects Renal Tubular Epithelial Cells of the Native Hibernator Syrian Hamster from Reoxygenation-Induced Cell Death"

_biology, 2019, doi:10.3390/biology8040074_

Round 1
Reviewer 1 Report
Eleftheriadis et al evaluated whether H2S-nuclear factor erythroid 2-like 2 (Nrf2)- antioxidant proteins axis protects RPTECs of the native hibernator Syrian hamster from cell death due to the reoxygenation that follows warm anoxia. This is an important finding that establishes how activation of H2S-Nrf2-antioxidant proteins axis following reperfusion rescues cells from reoxygenation-induced cell death. This work is well presented and will have a broader impact on the foundation of translating hibernation-based tolerance phenomenon in non-hibernating species for therapeutic intervention.
Comments:
The author investigated the activation of Nrf2 signaling pathway following reperfusion. Does activation of Nrf2 occur following reperfusion phase or during the anoxic phase since anoxia is also associated with cell death and triggers the self-repair mechanism? It would be nice to address the aspects of the anoxic phase associated with activation of H2S-Nrf2-antioxidant proteins axis. The author showed that following reoxygenation, there is a significant increase in the expression of Nrf2 (Figure 3 B, C) in RPTECs of the native hibernator Syrian hamster. Is the Nrf2 level cytoplasmic or nuclear? It has been well established that the newly synthesized Nrf2 released from the Keap1-Nrf2 complex translocates from the cytoplasm to the nucleus wherein, Nrf2 gets phosphorylated and sequentially binds to the antioxidant response element (ARE), a regulatory enhancer region within gene promoters. Why the author specifically looked into the level of Nrf2 but not p-Nrf2. Explain? Include what concentration of the epithelial cell media ingredients (such as epithelial cell growth supplement, fetal bovine serum, and antibiotics) are used for reference in the material and method section. Include the primary antibody dilution used for western blotting experiments. Page 4; Line 160: Missing comma after “H2S, Zinc acetate” Power analysis: Any a priori power analysis calculations were performed to determine the sample size that was used for detecting a significant difference within the groups.
Author Response
Response to the Reviewer 1
Firstly we would like to thank the reviewer since his/her kind comments encourage us to continue our research in the field of hibernation. Importantly, these comments helped us to improve the quality of our manuscript.
Does activation of Nrf2 occur following reperfusion phase or during the anoxic phase since anoxia is also associated with cell death and triggers the self-repair mechanism? It would be nice to address the aspects of the anoxic phase associated with activation of H2S-Nrf2-antioxidant proteins axis.
We assessed Nrf2 – antioxidant proteins axis under reoxygenation because in one of our previous studies we showed that in both mouse and hamster RPTECs, ROS production increases under reoxygenation, when oxygen is available, but not under anoxia. Thus, this axis could play a protective role only under reoxygenation. This point, along with the related citation, has been noted in the discussion section of the revised manuscript.
The author showed that following reoxygenation, there is a significant increase in the expression of Nrf2 (Figure 3 B, C) in RPTECs of the native hibernator Syrian hamster. Is the Nrf2 level cytoplasmic or nuclear? It has been well established that the newly synthesized Nrf2 released from the Keap1-Nrf2 complex translocates from the cytoplasm to the nucleus wherein, Nrf2 gets phosphorylated and sequentially binds to the antioxidant response element (ARE), a regulatory enhancer region within gene promoters. Why the author specifically looked into the level of Nrf2 but not p-Nrf2. Explain?
The Nrf2 corresponds to the total Nrf2 (cytoplasmic and nuclear), which in case of increased H2S production is released from Keap1, saved from proteasomal degradation and accumulated. Instead of measuring its activation in the nucleus, we assessed the level of several proteins whose gene transcription is known to be under the control of Nrf2. A note, along with the related citation, has been added in the discussion section of the revised manuscript.
Include what concentration of the epithelial cell media ingredients (such as epithelial cell growth supplement, fetal bovine serum, and antibiotics) are used for reference in the material and method section.
In the revised manuscript, all the above information is provided.
Include the primary antibody dilution used for western blotting experiments.
In the revised manuscript, all the above information is provided.
Page 4; Line 160: Missing comma after “H2S, Zinc acetate”
In the revised manuscript the type error was corrected.
Power analysis: Any a priori power analysis calculations were performed to determine the sample size that was used for detecting a significant difference within the groups.
We did not perform a priori power analysis calculations. We did nine experiments, and one-way ANOVA followed by Bonferroni’s correction test indicated the statistical significance of the results.
Reviewer 2 Report
The manuscript by Eleftheriadis el al examined the mechanism that protects hamster RPTEC from H/R injury as compared to mouse RPTECs. The authors hypothesized that hamster RPTECs produce more H2S compared to mouse cells and this could activate Nrf2 to translocate to nucleus to bind ARE to increase GST, SOD3 and Ferritin H thus causing less injury after H/R. overall, the study is very well done. The data presented is novel but requires more in depth in vivo analysis to demonstrate these finding to be true. The direct link between H2S and Nrf2 requires more studies, perhaps using siRNA for Nrf2 to demonstrate the direct involvement of Nrf2 in their cells lines. However, I think there is value in the finding. I have some minor concerns.
In figure 2, it is clear that mouse RPTEC have higher basal levels of CBS, CSE and 3-MST but these levels are not regulated after H/R. Does this mean that mouse RPTEC are more resistance to H/R compared to hamster cells? The entire western blots need to be shown.
In figure 3, I have the same concern. Is is clear that mouse cells have much higher levels of Nrf2 in control cells compared to hamster cells. This could be a reason why the levels of Nrf2 are not regulated to the same degree. This needs to be explained. The entire western blots need to be shown.
In figure 4 it seems that protein levels of GR, Ferrtin and xCT are much lower in mouse RPTEC thus this could mean that the mechanism the authors are testing is indirectly involved. SOD3 seems to be highly expressed but not regulated in mouse RPTEC as already pointed out for figure 2 and 3.
In figure 5, as shown it seems that mouse cells have higher cytotoxicity compared to hamster cells (this is contradictory to the western blot data in figure 2 and 3).
There are few typos line 312 (mahor to major)
Author Response
Firstly we would like to thank the reviewer for his/her kind and encouraging comments. Also, these comments helped us to clarify a certain and significant point of our study and improve our manuscript.
In figure 2, it is clear that mouse RPTEC have higher basal levels of CBS, CSE and 3-MST but these levels are not regulated after H/R. Does this mean that mouse RPTEC are more resistance to H/R compared to hamster cells? The entire western blots need to be shown. In figure 3, I have the same concern. Is is clear that mouse cells have much higher levels of Nrf2 in control cells compared to hamster cells. This could be a reason why the levels of Nrf2 are not regulated to the same degree. This needs to be explained. The entire western blots need to be shown. In figure 4 it seems that protein levels of GR, Ferrtin and xCT are much lower in mouse RPTEC thus this could mean that the mechanism the authors are testing is indirectly involved. SOD3 seems to be highly expressed but not regulated in mouse RPTEC as already pointed out for figure 2 and 3. In figure 5, as shown it seems that mouse cells have higher cytotoxicity compared to hamster cells (this is contradictory to the western blot data in figure 2 and 3).
A collective response
Mouse and hamster proteins were electrophoresed in different gels, and consequently, no direct comparison of protein levels could be performed. More importantly, this was done by purpose since even slight differences in the structure of a protein derived from different species can affect the affinity of the antibody used for the western blotting significantly. Thus, even if both mouse and hamster proteins were electrophoresed in the same gel, the direct comparison between them would be inaccurate. This point has been clarified in the methods section of the revised manuscript.
The type error has been corrected.